# A Geometric Perspective on Zero-Shot Variant Effect Prediction Across the Central Dogma

**César Miguel Valdez Córdova**
McGill University
Mila - Quebec AI Institute
Montréal, Quebec, Canada
`cesar.valdez@mila.quebec`

**Aaron Wenteler**
Queen Mary University of London
London, United Kingdom
`a.wenteler@qmul.ac.uk`

## Abstract

Log-likelihood ratios (LLR) have emerged as a standard probe of biological foundation models' ability to predict variant effects. However, it remains unclear whether the latent manifold of these models already encodes the relevant geometric structure of variant effects. We investigate this question across the central dogma by computing local intrinsic dimensionality (LID) around reference and alternative variant embeddings generated by EVO2 (DNA), ORTHRUS (RNA), and ESM3 (protein). This allows us to compare the geometric neighbourhoods of pathogenic and benign variants in each model's embedding space. On 105,224 ClinVar missense variants, the best-performing scoring method is modality-dependent: geometric scoring dominates for protein (ESM3 LID AUROC $= 0.738$ vs. LLR $= 0.629$), while likelihood-based scoring dominates for DNA (EVO2-7B LLR AUROC$= 0.878$). Dense per-layer analysis reveals three distinct information-processing regimes: stable geometric signal from the first layer onward (ESM3), monotonic buildup (ORTHRUS), and cyclic build-and-flush phases tied to the convolution–attention StripedHyena architecture (EVO2). The geometric signal persists after controlling for evolutionary conservation (57–61% retained) and is positive across all conservation quartiles, indicating that foundation models learn functional variant effects beyond what conservation scores capture.

## 1 Introduction

Predicting the functional effect of genetic variants has a long history. Early methods for predicting the functional variant effect leveraged evolutionary constraint directly, using conservation across homologous sequences or protein structure annotations to score substitutions (Ng & Henikoff, 2003; Adzhubei et al., 2010; Rentzsch et al., 2019). Deep generative models later showed that the fitness landscape could be learned directly from sequence alignments, scoring mutations by their log-likelihood ratio (LLR) with and without labelled data (Hopf et al., 2017; Riesselman et al., 2018; Frazer et al., 2021). Today, foundation models (FM) have been trained across the central dogma on large, unlabelled corpora of data for proteins (Meier et al., 2021; Lin et al., 2023; Hayes et al., 2025), RNA (Zou et al., 2024; Fradkin et al., 2025), and DNA (Avsec et al., 2021; Dalla-Torre et al., 2023; Nguyen et al., 2025; Avsec et al., 2026). These foundation models trained across the central dogma now provide the learned representations on which LLR prediction heads operate, making this the de facto approach to zero-shot pathogenicity scoring. Despite rich biological representations being learned through various self-supervised objectives, this approach reduces the model's learned representation to a single scalar, discarding representational detail that may be important to predicting variant effects.

In this work we move beyond LLR prediction and investigate whether the geometry of biological foundation model latent spaces is directly informative of variant effect. Rather than reducing a model's output to a scalar score, we ask what happens when we examine the full high-dimensional displacement that a variant induces in embedding space. This displacement encodes the complete effect of a variant as registered by the foundation model, yet has not been systematically exploited for pathogenicity prediction. Our central hypothesis is that the local geometry of the embedding manifold around a variant carries biological signal: pathogenic mutations, which violate hard biophysical

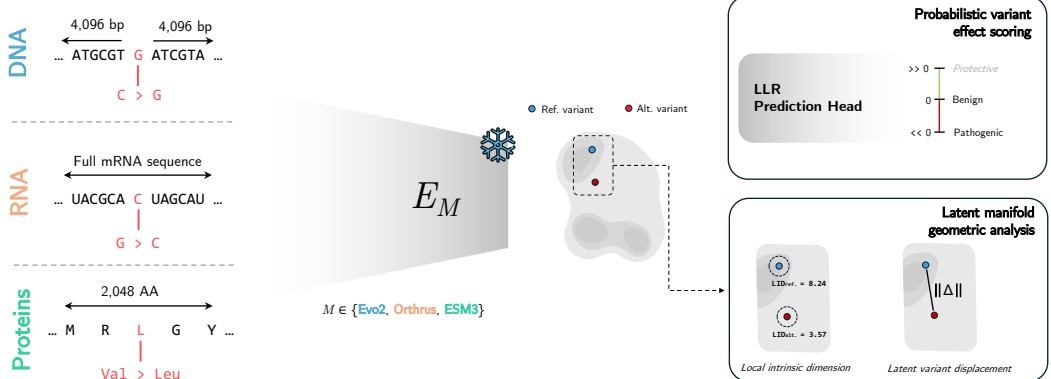

Figure 1: **Geometric probing of variant effect across the central dogma.** A single coding missense variant is represented at three levels: DNA, RNA, and protein, and passed through a frozen foundation model $E_M$ ($M \in \{$EVO2, ORTHRUS, ESM3$\}$). Reference (blue) and alternative (red) alleles are embedded into distinct positions on the learned manifold (centre). Two complementary analyses are performed. *Probabilistic variant effect scoring:* the model's LLR prediction head scores both alleles; negative values indicate predicted pathogenicity. *Latent manifold geometric analysis:* two properties of the embedding space are measured: the *local intrinsic dimensionality* (LID) of the neighbourhood around each variant's embedding, and the *latent variant displacement* (LVD, $\|\Delta\|$), the magnitude of the distance between the reference and alternative embeddings. Together, these probe whether the model's learned geometry encodes pathogenicity-relevant structure beyond what likelihood-based scoring captures.

constraints such as folding stability or active-site accessibility, should concentrate their displacements into low-dimensional subspaces, while benign substitutions scatter across a higher-dimensional space of tolerated variation. If this geometric structure exists, it would represent a source of variant effect information complementary to what LLR captures, one that is already present in the model's learned manifold. Conservation-based methods, while powerful, are limited to positions with sufficient phylogenetic depth and cannot capture constraints arising from epistatic interactions or context-dependent structural effects. Foundation model embeddings, trained on massive and diverse corpora of data through objectives that implicitly condition on full sequence context, optionally enriched by structure, should in principle provide a richer geometric substrate, capable of encoding the higher-order dependencies that position-level conservation scores, by construction, cannot represent.

We focus this exploration of multimodal variant effect modeling on missense variants. A single nucleotide variant (SNV) propagates through the central dogma: it alters a DNA codon, changes the mature mRNA transcript, and substitutes one amino acid in the translated protein. We model each level of biological information flow by a dedicated foundation model: EVO2 for DNA (Nguyen et al., 2025), ORTHRUS-MLM for RNA (Fradkin et al., 2025; Ian Shi et al., 2025), and ESM3 for protein (Hayes et al., 2025). By extracting geometric descriptors from the local latent manifolds of all three models, we evaluate the same set of ClinVar (Landrum et al., 2020) missense variants to determine if the geometric signature of pathogenicity is modality-dependent. Furthermore, we investigate whether manifold structure provides predictive insights in contexts where traditional probabilistic scoring falls short. Figure 1 illustrates this framework.

## 1.1 CONTRIBUTIONS

1. **Geometric scoring surfaces information that scalar methods miss.** On a shared set of ClinVar missense variants, LID-based scoring of ESM3 protein embeddings exceeds both LLR and LVD as a zero-shot pathogenicity discriminator (AUROC 0.738 vs 0.629 and 0.546, respectively), demonstrating that the local geometry of the embedding manifold encodes variant-effect information not captured by scalar summaries. The optimal scoring method is modality-dependent: DNA embeddings favor LLR, embeddings derived from mature mRNA favor LVD, and protein embeddings favor LID.

2. **Manifold geometry encodes functional constraint beyond conservation patterns.** Three properties of the geometric signal support its biological relevance: (i) it discriminates pathogenic from benign variants; (ii) it persists after controlling for phyloP conservation and is positive across all conservation quartiles; and (iii) it is specific to non-synonymous variation. Together, these indicate that foundation models learn functionally relevant semantics of variant effect beyond what conservation scores capture.

3. **Three modalities, three information-processing regimes.** Per-layer analysis reveals that variant effect information follows qualitatively distinct trajectories through model depth: stable from early layers (ESM3); monotonically increasing (ORTHRUS); and cyclically built and flushed in concert with the convolution–attention StripedHyena architecture (EVO2). This provides a mechanistic insight into how and where each model class best represents variant effects.

## 2 METHODS

### 2.1 DATA

We extract single-nucleotide variants (SNVs) from the full ClinVar VCF (Landrum et al., 2020) (GRCh38, downloaded January 2026), retaining variants with clinical significance in {`Pathogenic` (P), `Likely_pathogenic` (LP), `Benign` (B), `Likely_benign` (LB)}. All variants with P or LP annotations are considered pathogenic, whereas all variants with B or LB annotations are considered benign.

For missense variants, we compute the intersection of all model-specific filters (ESM3: protein length $\leq 2,048$ amino acids; ORTHRUS: RNA transcript all sequences accepted; EVO2: $\leq 8,193$ bp) and balance the majority class genome-wide (50/50 pathogenic/benign, seed=42), yielding 105,224 variants (52,612 per class). All three models encode the same variants, enabling direct cross-modality comparison.

DNA sequences are extracted from a local hg38 reference via pyfaidx ($\pm 4,096$ bp around each variant, 8,193 bp total). RNA and protein sequences are retrieved from the Ensembl REST API (Release 113).

### 2.2 FOUNDATION MODELS

**EVO2** (7B parameters): DNA foundation model with a StripedHyena2 (Ku et al., 2025) backbone, which interleaves three Hyena convolution operators (short FIR, medium FIR, long IIR) with attention layers across 32 blocks. Mean-pooled embeddings (4,096-dim) over 8,192-bp context windows. We extract 9 layers spanning 0–100% depth for per-layer analysis (Appendix D).

**ORTHRUS-MLM** (10M parameters, MLM checkpoint): RNA foundation model with Mamba SSM architecture. Mean-pooled embeddings (512-dim) from mature mRNA transcripts; all 6 layers are extracted for per-layer analysis.

**ESM3-SM** (1.4B parameters): Protein language model with masked prediction. Mean-pooled embeddings (1,536-dim, max 2,048 amino acids). We extract 11 layers spanning 0–100% depth for per-layer analysis.

### 2.3 LOCAL INTRINSIC DIMENSIONALITY

We measure the geometric structure of variant embedding neighborhoods using the MLE estimator of Local Intrinsic Dimensionality (LID) (Amsaleg et al., 2015). For each point $x_i$ with $k$-nearest neighbor distances $d_1, \ldots, d_k$:

$$\mathrm{LID}(x_i) = -\left( \frac{1}{k} \sum_{j=1}^{k} \log \frac{d_j}{d_k} \right)^{-1} \tag{1}$$

We use $k = 20$ with Euclidean distance. LID values that are non-finite, negative, or exceed 10,000 are excluded. Lower LID indicates fewer local degrees of freedom. We also compute Participation

Ratio (PR; variance-weighted effective dimension from SVD of the local neighbor matrix) and Tangent Space Approximation (TSA; hard rank of the local tangent plane). These metrics form a *sensitivity hierarchy* (Valdez Cordova et al., 2025): LID provides continuous per-sample estimates, PR aggregates local variance structure, and TSA applies a hard rank threshold, each successively coarser. We report LID as the primary metric; PR and TSA serve as confirmatory measures at decreasing resolution (Section 3.4). It is important to note that raw LID scores can only be interpreted within a modality, and not across them.

## 2.4 LATENT VARIANT DISPLACEMENT

We define the *latent variant displacement (LVD)* as the magnitude of the difference between mutant and wild-type mean-pooled embeddings:

$$\Delta = \|\bar{h}_\theta(\text{ALT.}) - \bar{h}_\theta(\text{REF.})\| \tag{2}$$

where $\bar{h}_\theta(x) = \frac{1}{L}\sum_{i=1}^{L} h_\theta^{(i)}(x)$ is the mean-pooled hidden state across all $L$ positions in the sequence. The LVD captures the full effect of a single-nucleotide variant as registered by the foundation model's learned representation. Larger values of this metric indicate that the variant moves the embedding further from wild-type in embedding space, which has been used as a proxy for evolutionary constraint in order to perform embedding-based annotation transfer (Heinzinger et al., 2022). All geometric analyses in this work, including LID estimation and the sensitivity hierarchy described below, operate on these LVD vectors.

## 2.5 LOG-LIKELIHOOD RATIO

The *log-likelihood ratio (LLR)* scores a variant by comparing the model's assigned probability to the reference and alternative alleles. The computation is architecture-dependent. For autoregressive models (EVO2), the LLR is read directly from the causal output logits:

$$\text{LLR} = \log p_\theta(x_{\text{ref}} \mid x_{<i}) - \log p_\theta(x_{\text{alt}} \mid x_{<i})$$

which we apply to EVO2 at the variant position. For masked language models, the variant position is masked and both alleles are scored under the surrounding context:

$$\text{LLR} = \log p_\theta(x_{\text{ref}} \mid x_{\setminus i}) - \log p_\theta(x_{\text{alt}} \mid x_{\setminus i})$$

which we apply to ESM3 via its sequence prediction head and to ORTHRUS via its MLM head at the mutated codon position. In all cases, the LLR is computed in the mutant sequence context. Positive values indicate that the model favours the reference allele; values near zero indicate no strong preference between alleles; and negative values indicate a preference for the alternative. Following Meier et al. (2021), we treat the LLR as a zero-shot pathogenicity score without any supervised training.

## 2.6 CONSERVATION BASELINE

We obtain per-position evolutionary conservation scores from the phyloP 100-way vertebrate track (Pollard et al., 2010) (UCSC hg38 bigWig). Positive scores indicate conservation; negative scores indicate acceleration. We compute AUROC using phyloP scores as a pathogenicity classifier to establish the conservation baseline.

## 2.7 STATISTICAL ANALYSIS

Group comparisons (P vs B LID distributions) use the Mann-Whitney U test (two-sided). Correlations are Spearman rank correlations. We control for confounders using partial Spearman correlation:

$$\rho_{XY \cdot Z} = \frac{\rho_{XY} - \rho_{XZ}\,\rho_{YZ}}{\sqrt{(1 - \rho_{XZ}^2)(1 - \rho_{YZ}^2)}} \tag{3}$$

where $X$ represents the LID, $Y$ the pathogenicity label, and $Z$ the phyloP conservation score. We additionally perform stratified analysis by splitting variants into phyloP quartiles and computing the LID gap within each stratum.

## 3 RESULTS

### 3.1 ZERO-SHOT VARIANT SCORING ACROSS MODALITIES

Figure 2A compares three zero-shot scoring methods across all three modalities on the shared set of 105,224 missense variants, using the best intermediate layer for EVO2 (`blocks.24`), and the final layer for both ESM3 and ORTHRUS. The optimal scoring method is modality-dependent. ESM3 protein embeddings achieve the highest LID AUROC (0.738), substantially exceeding both its LLR (0.629) and LVD (0.546). The EVO2 DNA FM achieves the highest LLR (0.878) and LVD (0.809) scores, far exceeding ESM3 on these metrics, yet its LID (0.667) remains below ESM3's. For RNA, LVD seems to separate pathogenic and benign variants best, with an LVD score of (0.619).

### 3.2 PER-LAYER DYNAMICS ACROSS MODALITIES

To understand how pathogenicity information is processed through the models' layers, we extract embeddings from dense per-layer sampling and compute LID-based AUROC at each layer (Figure 2B). Three distinct patterns emerge:

**ESM3 (stable):** The geometric signal is present from the very first transformer block (LID AUROC = 0.722 at layer 0) and remains remarkably stable across all 48 layers, reaching 0.738 at the final layer, a variation of only 0.016 across the full depth. LVD AUROC is flat at ∼0.55 throughout, never exceeding 0.556. The fact that LID signal is already strong at the input projection, before any attention has been applied, suggests that ESM3's learned token embeddings already encode constraint-relevant structure.

**ORTHRUS (building):** Signal increases monotonically from near-chance at layer 0 (0.507) to its maximum at the final layer (0.586). The largest jump occurs between layers 3 and 4 (0.533 → 0.582), suggesting a phase transition in how the Mamba state-space model encodes variant effects.

**EVO2 (cycling):** Dense per-layer analysis reveals a cyclic build-and-flush pattern tied to the Striped-Hyena2 architecture. LVD AUROC rises from 0.584 (block 0) to a peak of 0.768 (block 8), then gradually decays to chance by the final block (0.500). LID, however, follows a strikingly different trajectory: it oscillates between near-chance and weak signal in the early and middle layers (blocks 0–16), with direction reversals at blocks 0, 8, and 10 where *higher* LID associates with pathogenicity (the opposite of the standard direction). The dips at blocks 6 and 16 (LID AUROC ≈ 0.50) correspond to long-range IIR convolution layers, which integrate over the full sequence

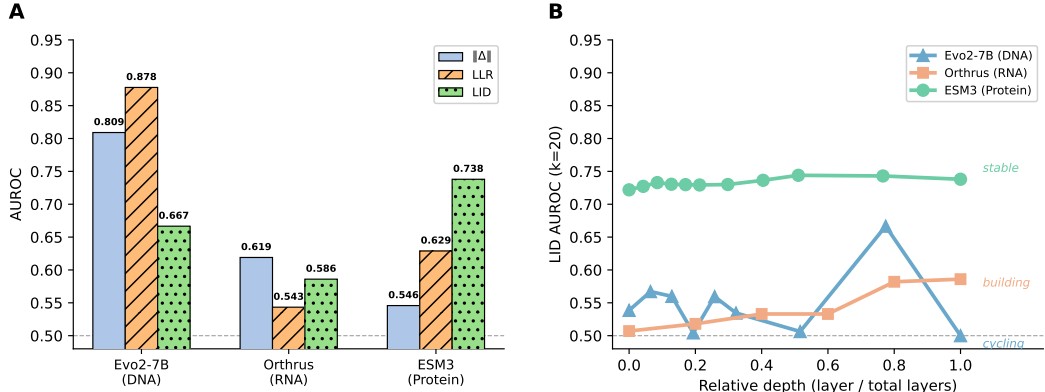

Figure 2: **Cross-modality pathogenicity scoring.** (A) Three zero-shot scoring methods: latent variant displacement ($\|\Delta\|$), log-likelihood ratio (LLR), and LID-based geometric scoring, across protein (ESM3), RNA (ORTHRUS), and DNA EVO2) modalities. (B) Dense per-layer LID AUROC across model depth reveals three distinct patterns: ESM3 is stable from the first layer (0.722) through the last (0.738); ORTHRUS builds monotonically with a phase transition at layer 4; EVO2 cycles through build-and-flush phases tied to its convolution–attention architecture, peaking at the attention layer at 75% depth before collapsing by the final layer. Dashed line: chance (AUROC = 0.5).

context and appear to dilute the single-variant signal across the representation. Only at `blocks.24`, an attention layer at 75% depth, does a strong, correctly-oriented geometric signal emerge (LID AUROC $= 0.667$, gap $= +16.88$, $p < 10^{-300}$). This signal is then flushed by the subsequent StripedHyena convolution layers (blocks 25–30), leaving the final attention layer (`blocks.31`) numerically degenerate ($\|\Delta\| \sim 10^{-17}$, all metrics at chance).

### 3.3 GEOMETRIC–MAGNITUDE DISSOCIATION AND VARIANT SPECIFICITY

For ESM3 protein embeddings, the local geometry of the embedding manifold is far more predictive of pathogenicity than the size of the displacement itself: LID AUROC exceeds LVD AUROC by 19 points (0.738 vs. 0.546). Pathogenic variants occupy systematically lower-dimensional neighbourhoods (median LID $= 0.12$) compared to benign variants (median LID $= 0.77$), indicating that the two classes land in geometrically distinct regions of the manifold even when their displacement magnitudes are comparable.

This geometric signal is specific to missense variants. In EVO2 DNA embeddings, synonymous, splice, and UTR variants all produce non-zero displacements yet show no significant LID gap between pathogenic and benign classes. Since missense variants are the only class that alters the translated protein product, this specificity is consistent with protein-level functional constraint as a driver of the geometric separation. This suggests that a DNA-level model, trained without any protein supervision, learns latent geometric structures that tracks the functional consequences of amino acid substitution.

### 3.4 METRIC SENSITIVITY HIERARCHY

Table 1 reports PR and TSA alongside LID at $k = 20$. For ESM3, where the per-sample geometric signal is strongest, all three metrics agree: PR gap $= +1.41$ and TSA gap $= +3.0$ (both $p < 10^{-10}$). EVO2 `blocks.24` presents a more revealing case: a large LID gap ($+16.88$) and PR gap ($+2.43$) coexist with zero TSA gap, indicating that pathogenic and benign neighbourhoods share the same number of active dimensions yet differ in how variance is distributed across them. In other words, pathogenic and benign variants do not reside on manifolds of a different dimension, but their local neighbourhoods distribute variance across similarly ranked subspaces in systematically different ways. ORTHRUS exhibits the same pattern: a strong LID signal ($+6.71$) and modest PR gap ($+0.35$) alongside zero TSA gap, indicating that the graded geometric separation exists for foundation models trained on DNA as well as mature mRNA.

Table 1: Geometric metric sensitivity hierarchy at $k = 20$. As signal strength decreases across modalities, coarser metrics (TSA, then PR) lose sensitivity while LID continues to detect geometric separation. All nonzero gaps significant at $p < 10^{-10}$ (Mann-Whitney U) given $N = 105{,}224$; we caution that significance at this sample size reflects statistical power, not necessarily effect magnitude. *Numerically degenerate: final-layer perturbations have $\|\Delta\| \sim 10^{-17}$, making SVD-based metrics (PR, TSA) unreliable.

| Model | LID gap | PR gap | TSA gap |
|---|---|---|---|
| ESM3 (Protein) | $+0.65$ | $+1.41$ | $+3.0$ |
| EVO2 `blocks.24` (DNA) | $+16.88$ | $+2.43$ | $0.0$ |
| ORTHRUS (RNA) | $+6.71$ | $+0.35$ | $0.0$ |

### 3.5 RELATIONSHIP TO EVOLUTIONARY CONSERVATION

A natural concern is whether the geometric signal simply recapitulates evolutionary conservation. We address this through partial Spearman correlation controlling for phyloP 100-way vertebrate conservation scores (Pollard et al., 2010) and stratified analysis by conservation quartile (Table 2).

Across all three modalities, approximately 57–61% of the raw LID-pathogenicity correlation is retained after controlling for phyloP. ESM3's raw correlation ($\rho = -0.412$) drops to $-0.245$, EVO2 `blocks.24` drops from $-0.289$ to $-0.175$, and ORTHRUS from $-0.148$ to $-0.084$. The consistency of the residual across modalities suggests that all three model classes learn geometric structure that is partially but not fully explained by position-level conservation.

The LID gap is positive across all four conservation quartiles for every modality (Table 2), confirming that the geometric signal exists at every level of conservation. However, the quartile trends reveal a striking modality dependence. ESM3's LID gap *decreases* with increasing conservation ($+0.71 \rightarrow +0.20$), suggesting that protein-level geometric structure is most informative precisely where conservation alone is weakest. EVO2 shows the opposite trend ($+10.6 \rightarrow +15.6$), with the strongest geometric separation at the most conserved positions. This inversion is consistent with the two models capturing complementary aspects of constraint: ESM3 encodes protein-specific structural and functional features that go beyond conservation, while EVO2's geometric signal may reflect sequence-level patterns of purifying selection that amplify at highly conserved loci.

Table 2: PhyloP confounder control across modalities. Partial Spearman correlation controls for phyloP conservation. EVO2 uses intermediate layer `blocks.24`.

| Model | $\rho_{\textbf{raw}}$ | $\rho_{\textbf{|PhyloP}}$ | Retained |
|---|---|---|---|
| ESM3 (Protein) | $-0.412$ | $-0.245$ | 59% |
| EVO2 B.24 (DNA) | $-0.289$ | $-0.175$ | 61% |
| ORTHRUS (RNA) | $-0.148$ | $-0.084$ | 57% |

| Model | Q1 (low) | Q2 | Q3 | Q4 (high) |
|---|---|---|---|---|
| ESM3 (Protein) | $+0.71$ | $+0.56$ | $+0.29$ | $+0.20$ |
| EVO2 B.24 (DNA) | $+10.6$ | $+13.6$ | $+13.0$ | $+15.6$ |
| ORTHRUS (RNA) | $+1.67$ | $+4.81$ | $+5.02$ | $+6.60$ |

## 3.6 GENE-LEVEL ROBUSTNESS

To ensure that the LID signal is not merely an artifact of variant representation bias in ClinVar (small set of genes contributing disproportionally many variants to the evaluation), we investigated whether our performance was driven by a small subset of well-annotated disease genes. The dataset comprises 105,224 variants distributed across 12,170 genes. We conducted a sensitivity analysis by iteratively excluding the top-$N$ genes by variant count and recomputing the AUROC.

The ten most frequent genes account for 5.2% of the total observed variation. Removing these genes resulted in a marginal AUROC decrease from $0.744$ to $0.730$. Even after excluding the top 50 genes, which represent 13.8% of the variation, the AUROC remained robust at $0.713$. These results indicate that the geometric signal is a genome-wide phenomenon rather than a bias localized to a few dominant loci. Similar stability was observed for LLR and LDV scores (see Appendix H).

## 4 DISCUSSION

### 4.1 TOWARD A GEOMETRIC INTERPRETATION

One hypothesis consistent with our observations is that foundation models learn *constraint manifolds*: pathogenic variants may violate a small number of strong constraints (folding, active site geometry, stability), concentrating pathogenic variation in a low-dimensional subspace, while benign variants scatter across a higher-dimensional space of tolerated substitutions. The cross-modality hierarchy is consistent with this view: ESM3, trained directly on protein sequences, shows the strongest geometric signal; ORTHRUS captures weaker constraint geometry through RNA; EVO2 encodes geometric information at intermediate layers but loses it by the final layer. While we present here some compelling results to support the *constraint manifold* hypothesis, we emphasize that it is not yet conclusively confirmed. Future work linking LID to structural features (e.g. buried vs. surface residues, active site proximity, secondary structure context) is needed to establish confirmation.

The per-layer patterns we observed across modalities provide empirical constraints on this hypothesis. ESM3's geometric signal is present from the very first layer ($0.722$) and varies by only $0.016$ across all 48 transformer blocks, suggesting that constraint-relevant structure is encoded in the token embeddings and preserved by the model rather than constructed through depth. For mature mRNA, ORTHRUS progressively refines its constraint manifold as it progresses through successive Mamba–MLM layers.

EVO2's pattern is the most architecturally revealing. Dense per-layer analysis shows that the LID signal oscillates rather than declines monotonically, with dips at long-range IIR convolution layers (which integrate over the full 8k bp context window, diluting single-nucleotide variant representations) and partial recoveries at short-range FIR layers (which re-access local context). Only the attention layers can re-aggregate this information globally; the peak at `blocks.24` (the penultimate attention layer at 75% depth) reflects the optimal balance between sufficient feature depth and proximity to the final Hyena cascade (blocks 25–30) that flushes the signal through long-range convolutional operators. The build-and-flush pattern persists across model scales (Appendix B), with dramatically stronger intermediate signal at 7B (LID gap $+16.88$), confirming that the compression is an intrinsic property of the causal convolution architecture rather than a capacity limitation.

Beyond architectural dynamics, the geometric signal also appears to capture information complementary to both conservation and likelihood-based scoring. Both ESM3 and EVO2-7B show partial overlap with phyloP (Table 2), yet the $\sim$60% residual in both modalities demonstrates constraint structure beyond conservation. The weak correlation between LID and LLR ($\rho = -0.12$ for ESM3) further suggests that geometric and probabilistic scoring capture distinct aspects of constraint. A logistic regression combination analysis (5-fold stratified CV) provides a preliminary test: phyloP alone achieves AUROC $= 0.895$, while adding LID and LLR yields $0.901$ ($+0.006$). The practical improvement is modest, underscoring that the primary contribution of this work is analytical—understanding *what* foundation model embeddings encode geometrically—rather than proposing a new variant effect classifier.

## 4.2 LIMITATIONS

Several limitations should be noted. First, ClinVar over-represents Mendelian disease variants and well-studied genes; generalization to rare variants and under-studied genes remains to be established. Second, we use $k = 20$ as the primary neighborhood size for LID estimation. ESM3's LID gap increases monotonically with $k$ ($+0.38$ at $k = 10$ to $+2.64$ at $k = 100$), confirming robustness; however, 11.9% of protein samples have degenerate ($-\infty$) LID at $k = 20$ due to exact duplicate perturbation vectors (zero $k$NN distances), decreasing to 2.4% at $k = 100$. These duplicates likely arise from short proteins where mean-pooling collapses distinct single-residue mutations to identical representations, or from mutations at positions where ESM3's learned representation is locally flat. DNA embeddings show 0% filtering at all $k$. Third, ORTHRUS RNA embeddings used a masked language model checkpoint (512 dimensional) that differs from the base pre-trained model (256 dimensional); the MLM head enables LLR computation but introduces a potential confound. Fourth, we use EVO2-7B rather than the larger EVO2-40B used in the original paper's supervised evaluations (see Appendix A); results with the smaller EVO2-1B are reported in Appendix B, where the same architectural pattern is visible but all metrics are near chance. Finally, we focus our analyses here on missense variants only, excluding a large portion of the types of genetic variation frequent in nature.

## 5 CONCLUSION

Foundation models encode variant pathogenicity through geometric constraint structure that standard scoring methods partially miss. On 105,224 ClinVar missense variants, the optimal scoring method is modality-dependent: ESM3 protein achieves the highest LID AUROC (0.738), while EVO2-7B DNA achieves the highest LLR (0.878) and perturbation magnitude (0.809) at its best intermediate layer. The geometric signal persists after controlling for evolutionary conservation (57–61% retained depending on modality), is missense-specific, and is positive across all conservation quartiles.

Dense per-layer analysis reveals how models process variant information: ESM3 encodes stable geometric signal from the very first layer (0.722) onward, ORTHRUS builds signal monotonically through depth, and EVO2's signal cycles through build-and-flush phases tied to its convolution–attention architecture, peaking at an attention layer (LID AUROC $= 0.667$) before collapsing by the final layer. The final-layer collapse persists at the smaller 1B scale (Appendix B), suggesting that it is an intrinsic property of the causal convolution architecture. These distinct patterns suggest that geometric analysis can extract variant-effect information even from models whose standard outputs fail to discriminate pathogenicity. Whether this geometric structure reflects true constraint manifolds or a different underlying mechanism remains an open question for future work.

## Meaningfulness Statement

As biological foundation models scale, benchmark performance alone cannot distinguish whether failures reflect data limitations, architectural choices, or training objectives, particularly when benchmarks are proxies for the molecular phenomena we actually seek to model. Geometric analysis of learned representations offers a diagnostic trace: by examining *how* models encode variant effects across their depth and across modalities, we can identify where biologically meaningful structure is learned, where it is destroyed, and why. Our finding that EVO2 learns and then erases pathogenicity-relevant geometry illustrates precisely this: a failure invisible to standard evaluation but revealed by tracing the manifold. Understanding what representations encode is prerequisite to designing what they should.

## Acknowledgements

The authors would like to thank G. Gonzalez for his helpful feedback regarding zero-shot variant effect prediction with EVO2. The authors would also like to thank P. Fradkin for providing the weights to ORTHRUS-MLM and his feedback regarding the model. The geometric analyses in this work were built on `manylatents`[1] and its extension `manylatents-omics`[2], part of the Latent Reasoning Works open-source ecosystem for geometric analysis of learned representations. Meaningful structure is learned through many latents; we hope this work helps trace where.

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

APPENDIX

## A  COMPARISON WITH EVO2 SUPERVISED RESULTS

Our EVO2-7B zero-shot results differ from Nguyen et al. (2025)'s supervised AUROC $= 0.95$ on BRCA1. Key methodological differences: (1) **Model scale**: we use EVO2-7B (7 billion parameters, 32 blocks, hidden dimension 4096) vs their 40B. (2) **Supervised vs zero-shot**: they train a 3-layer classifier on labeled variants; we use no labels. (3) **Gene-specific vs genome-wide**: they evaluate on BRCA1 only; we evaluate on 105,224 variants spanning 16,436 genes. (4) **Feature engineering**: they use narrow 128-nt windows at block 20 with 4 sequence views (32,768-dim features); we mean-pool over 8k-bp context, diluting the single-SNV signal across $\sim$8,192 positions.

Notably, their finding that block 20 (intermediate depth) yields peak supervised performance is consistent with our per-layer analysis. Block 20 is a long-range IIR convolution layer; our dense analysis shows that the best *unsupervised* geometric signal comes from `blocks.24` (the nearest attention layer), suggesting that attention-based re-aggregation is particularly important for zero-shot geometric scoring. With EVO2-7B (32 blocks, 4096-dim), the peak intermediate layer achieves LID AUROC $= 0.667$ and LLR AUROC $= 0.878$, while the final layer remains at chance. We report EVO2-1B final-layer results in Section B; all metrics are near chance, consistent with the build-and-flush pattern but without sufficient model capacity for strong intermediate signal. The remaining gap with Nguyen et al. (2025) likely reflects model scale (7B vs 40B) and task-specific feature engineering (narrow 128-nt windows vs full-context mean pooling).

## B  EVO2-1B RESULTS

For comparison, we report zero-shot results using the smaller EVO2-1B model (1 billion parameters, 25 StripedHyena blocks, 1920-dimensional embeddings). All metrics are at or below chance at the final layer, demonstrating that the build-and-flush architecture produces degenerate final-layer representations regardless of scale. These results serve as a cautionary baseline: 1B-scale DNA foundation models should not be expected to yield meaningful zero-shot pathogenicity signal under mean-pooled geometric analysis.

Table 3: EVO2-1B zero-shot pathogenicity scoring on 105,224 missense variants.

| Model | Dim | $\|\Delta\|$ | LLR | LID ($k\,{=}\,20$) |
|---|---|---|---|---|
| EVO2-1B (final) | 1920 | 0.503 | 0.500 | 0.498 |

## C  ZERO-SHOT SCORING PROCEDURES

All three scoring methods produce a single scalar per variant; AUROC measures how well each metric's ranking aligns with ClinVar labels. **No method uses labeled data for training.**

**Perturbation magnitude.**  $\|\Delta\|_2 = \|\bar{h}_\theta(\mathrm{MUT}) - \bar{h}_\theta(\mathrm{WT})\|_2$. Higher magnitude is treated as more likely pathogenic.

**Log-likelihood ratio.**  For autoregressive models (EVO2), LLR $= \log p_\theta(x_{\mathrm{ref}} \mid x_{<i}) - \log p_\theta(x_{\mathrm{alt}} \mid x_{<i})$ directly from the causal output logits. For masked language models (ORTHRUS, ESM3), the variant position is masked and both alleles scored: LLR $= \log p_\theta(x_{\mathrm{ref}} \mid x_{\setminus i}) - \log p_\theta(x_{\mathrm{alt}} \mid x_{\setminus i})$. All LLR is computed in the mutant sequence context.

**LID-based scoring.**  Per-sample LID at $k = 20$ over all 105,224 perturbation vectors (FAISS-GPU $k$NN). AUROC$_{\mathrm{LID}} = P(\mathrm{LID}_{\mathrm{path}} < \mathrm{LID}_{\mathrm{benign}})$—pathogenic variants in lower-dimensional regions score higher. Raw LID values are not comparable across modalities; AUROC normalizes by measuring relative ranking within each modality.

## D  PER-LAYER EXTRACTION METHODOLOGY

We extract intermediate-layer embeddings at dense intervals across all three models.

**ESM3 (48 transformer blocks).**  We register a `forward_hook` on the `TransformerStack` to intercept hidden states at layers 0, 2, 4, 6, 8, 10, 14, 19, 24, 36, and 47 (11 layers spanning 0–100% depth) without modifying library code, capturing only requested layers to limit memory. Each hidden state is mean-pooled with the same attention mask as the final layer.

**ORTHRUS (6 Mamba blocks).**  The `MixerModel.representation(capture_layers=...)` interface exposes all 6 layers directly. We extract all layers (0–5). We use the MLM-6-track checkpoint (512-dim embeddings, vs 256 for the base model) which provides a `SequenceProjectionHead` for LLR computation.

**EVO2 (32 StripedHyena blocks).**  Native per-layer extraction via the `layer_names` parameter at 9 layers: `blocks.0, 2, 4, 6, 8, 10` (0–32% depth), `blocks.16` (50%), `blocks.24` (75%), and `blocks.31` (final). The block types cycle as HCS (short FIR) $\rightarrow$ HCM (medium FIR) $\rightarrow$ HCL (long IIR) $\rightarrow$ attention, with attention at blocks 3, 10, 17, 24, and 31. LLR is computed autoregressively from position $i-1$ logits.

## E  LID SENSITIVITY TO NEIGHBORHOOD SIZE

Table 4 shows LID gap across $k \in \{10, 20, 50, 100\}$ for each modality. ESM3 protein shows monotonically increasing gaps, indicating that pathogenic variants reside in systematically lower-dimensional neighborhoods at all scales. EVO2-7B `blocks.24` shows the largest gaps at small $k$ ($+23.93$ at $k=10$), decreasing monotonically to $+5.18$ at $k=100$. ORTHRUS RNA maintains large gaps across all $k$ values with a slight non-monotonic pattern. The EVO2 final layer shows no signal at any $k$.

Table 4: LID gap (pathogenic $-$ benign median) across neighborhood sizes $k$. ESM3 signal strengthens with $k$; EVO2-7B `blocks.24` signal weakens but remains highly significant; EVO2 final layer shows no signal at any $k$.

| Model / Layer | $k=10$ | $k=20$ | $k=50$ | $k=100$ |
|---|---|---|---|---|
| ESM3 (Protein, final) | $+0.38$ | $+0.65$ | $+1.46$ | $+2.64$ |
| EVO2-7B `blocks.24` | $+23.93$ | $+16.88$ | $+9.82$ | $+5.18$ |
| ORTHRUS (RNA, final) | $+8.29$ | $+6.71$ | $+5.98$ | $+6.27$ |
| EVO2-7B (DNA, final) | $\approx 0$ | $\approx 0$ | $\approx 0$ | $\approx 0$ |

## F  EFFECT SIZES

Per-variant effect sizes are modest (Table 5), consistent with the expectation that individual variants produce subtle geometric displacements; the discriminative signal emerges from consistent distributional separation across the full variant population. Raw LID gaps are not comparable across modalities due to differing LID scales; AUROC normalizes this and is the primary cross-model comparator.

## G  FEATURE COMBINATION

We evaluate feature complementarity using logistic regression with 5-fold stratified cross-validation on the variants with finite LID, LLR, and phyloP scores (88.1% of 105,224; the remainder have degenerate LID, see Section 4.2). The excluded variants are heavily enriched for pathogenic (82.9% pathogenic vs. 50% base rate), so this subset is slightly conservative, implying the degenerate variants are disproportionately constrained positions. PhyloP alone achieves AUROC $= 0.895 \pm 0.001$ on this subset (vs. 0.894 on the full set, confirming minimal selection bias). Adding ESM3 LID yields 0.898

Table 5: Standardized effect sizes (Cohen's $d$) for LID separation between pathogenic and benign variants at $k = 20$. All effects are "small" by conventional thresholds ($d < 0.5$), reflecting the subtlety of individual geometric displacements.

| Model | LID gap | Cohen's $d$ | AUROC |
|---|---|---|---|
| ESM3 (Protein) | +0.65 | 0.158 | 0.738 |
| EVO2-7B `blocks.24` | +16.88 | 0.356 | 0.667 |
| ORTHRUS (RNA) | +6.71 | 0.115 | 0.586 |

(+0.003); adding LLR yields 0.899 (+0.004); combining all three achieves 0.901 (+0.006). The improvement is modest, consistent with the moderate correlation between LID and phyloP ($\rho = 0.37$). The weak LID-LLR correlation ($\rho = 0.13$) confirms that geometric and probabilistic scoring capture distinct information, though a linear model may not fully exploit this complementarity.

## H  GENE-LEVEL ROBUSTNESS

The 105,224 variants span 12,170 unique genes. Table 6 shows the effect of excluding well-annotated genes on all three scoring methods. LID AUROC degrades gracefully, confirming that the signal is not driven by a few dominant genes. The top 10 genes, which we also refer to in the main text, are: *LDLR, COL4A5, GLA, COL3A1, COL1A2, COL1A1, COL2A1, PAH, BRCA1,* and *MYH7*.

Table 6: AUROC under gene exclusion (ESM3 protein). Top-$N$ genes by variant count are removed; remaining variants are scored. The LID signal degrades gradually; magnitude and LLR are stable. LID values use exact $k$-NN and differ slightly from the approximate FAISS values in the main text (0.744 vs 0.738); relative comparisons are unaffected.

| Condition | $N$ variants | LID | $\|\Delta\|$ | LLR |
|---|---|---|---|---|
| Full set | 105,224 | 0.744 | 0.546 | 0.629 |
| Excl. top 10 | 99,725 | 0.730 | 0.559 | 0.630 |
| Excl. top 20 | 96,696 | 0.725 | 0.561 | 0.628 |
| Excl. top 50 | 90,722 | 0.713 | 0.561 | 0.624 |

## I  DATA PROCESSING NOTES

**RNA variant injection.**  ORTHRUS requires mature mRNA sequences. To construct mutant RNA sequences, we identify the variant position within the coding sequence using codon tables and the amino acid change annotation. For variants on the reverse strand, reverse complement logic is applied. Of the 105,224 missense variants, 7,712 (7.3%) could not be injected into their RNA transcripts, typically because the annotated amino acid change did not match any position in the retrieved transcript. These variants receive zero-valued perturbation vectors ($\Delta = 0$) and are included in the analysis; excluding them does not materially change any reported AUROC.

**PhyloP conservation baseline.**  We compute per-position PhyloP 100-way vertebrate conservation scores from the UCSC bigWig file (hg38.phyloP100way.bw) at each variant's genomic coordinate. PhyloP AUROC = 0.894 on the shared 105,224 missense set establishes a strong conservation-based baseline. The LID signal's relationship to conservation is modality-dependent: ESM3 retains 59% of the raw effect after controlling for phyloP ($\rho_{partial} = -0.245$ vs. $\rho_{raw} = -0.412$), EVO2-7B retains 61% ($-0.175$ vs. $-0.289$), and ORTHRUS retains 57% ($-0.084$ vs. $-0.148$). The LID gap is positive across all PhyloP quartiles for every modality with signal (see Table 2).

