# OpenReview forum: "A Geometric Perspective on Zero-Shot Variant Effect Prediction Across the Central Dogma"
_ICLR.cc/2026/Workshop/LMRL — ICLR 2026 Workshop LMRL Poster_

### Official Review · Reviewer_tb7o · 2026-02-24
**Novel Perspective, Limited Predictive Gains**

**Rating:** 7
**Confidence:** 3

**Review:**

# Workshop Fit

Good fit. The paper probes three biological foundation models across the central dogma (DNA, RNA, protein) and asks whether embedding geometry encodes functional constraint.

## Summary

The paper investigates whether biological FMs encode different geometric signatures for pathogenic versus benign variants, beyond the standard log-likelihood ratio (LLR). The authors apply Local Intrinsic Dimensionality (LID) and Latent Variant Displacement (LVD) to characterise the embedding perturbation induced by a variant, tested across EVO2-7B (DNA), ORTHRUS (RNA), and ESM3 (protein) on 105,224 ClinVar missense variants, with PhyloP conservation as a baseline and control. Key results: LID outperforms LLR and LVD for ESM3, while for EVO2 LLR substantially outperforms LID, and for ORTHRUS the two are close. A per-layer analysis identifies three dynamics: stable from early layers (ESM3), monotonically accumulating (ORTHRUS), and cyclical tied to architecture (EVO2). The LID signal retains the majority of its discriminative power after partialling out PhyloP. However, an ensemble of PhyloP, LID, and LLR yields no improvement over PhyloP alone.

## Evaluation

**Originality.** Applying LID to biological embedding perturbations as a pathogenicity signal is, to the reviewer's knowledge, novel.

**Quality.** The PhyloP control is well-executed, the k sensitivity scan and 5-fold CV are appropriate, and the authors are transparent about limitations. However zero-shot claim is only partially true or needs clarification.

**Clarity.** Well-written.

**Significance.** The detailed cross-modality analysis of embedding geometry, particularly the per-layer dynamics and the partial independence from conservation, is a genuine contribution to understanding what biological FMs encode, independent of whether LID/LVD outperforms the baseline. As a predictive tool, however, the gains are modest: LID and LVD do not improve meaningfully over PhyloP alone, and the best ensemble adds only +0.006.

---

## Pros

- LID and LVD as geometric characterisations of variant-induced embedding perturbations are novel scores; the cross-modality framing (DNA→RNA→protein) makes the analysis especially interesting.
- The per-layer dynamics analysis is architecturally informative
- The PhyloP partial-correlation control is and important control; retaining the majority of the LID signal after controlling for conservation is one of the stronger quantitative results.
- The authors transparently communicate limitations (mean-pooling, k choice, model size, checkpoint mismatch), and provide a k sensitivity scan and 5-fold CV.

---

## Cons

**1. EVO2 layer selection violates the zero-shot premise.** The optimal layer is identified by scanning AUROC over all EVO2 layers using ClinVar labels, supervised selection, not zero-shot. Without a label-free criterion, and without evidence that block 24 is consistently optimal across different variant subsets, other authors cannot apply EVO2 LID in a zero-shot fashion. The result remains tied to the specific labels used to identify it.

**3. Bootstrap CIs are missing for the complementarity claim.** The combined conditions (LID+PhyloP, all three) are reported as point estimates only. Whether the gains over PhyloP are statistically distinguishable from zero cannot be assessed.

**4. No comparison to specialised variant effect predictors.** The paper does not benchmark against tools such as AlphaMissense or EVE, making it difficult to contextualise the practical value of LID-based scoring.

**5. Generalisability of the observed patterns is unestablished.** The entire analysis rests on ClinVar missense variants, and the authors themselves acknowledge that replication on independent variant sets would be important. This is not a minor caveat: the per-layer dynamics and the LID signal are the paper's central findings, and it is crucial to know whether these patterns are specific to ClinVar or reflect a general property of how biological FMs encode functional constraint. Without at least one replication cohort, the findings remain exploratory observations on a single dataset.

## Questions

1. Is there a simple label-free criterion for selecting the optimal EVO2 layer, like variance of the score across layers?
2. The authors acknowledge that mean-pooling dilutes the single-residue signal. Is it possible to compute LID at the mutated position rather than averaging over the full sequence, and if so have the authors tried this? At least for EVO2 that should be feasible?
3. Can you provide bootstrap CIs for the combined conditions and a significance test for the improvement over PhyloP?
4. How does LID compare to AlphaMissense or EVE on the same variant set?

---

### Official Review · Reviewer_KqBZ · 2026-02-24
**This paper studies whether the geometry of foundation model embedding spaces carries variant-effect signal beyond standard LLR scoring. The authors compare three biological foundation models across the central dogma (EVO2 for DNA, ORTHRUS for RNA, ESM3 for protein) on a shared ClinVar missense benchmark, using geometric metrics such as LID, PR and TSA. The main claims are that (i) the best zero-shot scoring method is modality-dependent (e.g., LID strongest for protein/ESM3, LLR strongest for DNA/EVO2), (ii) the geometric signal persists after controlling for conservation (phyloP), and (iii) per-layer analysis reveals distinct information-processing regimes across architectures.    I found the paper novel, well-motivated at a high level, and well-executed as a workshop paper, with strong analysis and clear presentation overall. The central-dogma comparison is especially interesting.**

**Rating:** 8
**Confidence:** 4

**Review:**

This paper studies whether local geometry of foundation-model embeddings (via LID/PR/TSA on latent variant displacement) provides pathogenicity signal beyond LLR and conservation, using a central-dogma comparison across EVO2 (DNA), ORTHRUS (RNA), and ESM3 (protein) on ClinVar missense variants. The paper is novel, thorough, and well written overall. I found it a good workshop contribution.


Strengths

1) Novelty

The paper asks a good question: whether variant-effect information is present in the latent geometry of biological FMs, rather than only in scalar LLR readouts. This is a meaningful and timely contribution, especially given the widespread use of LLR as the default zero-shot probe.

2) Central dogma frame

The DNA/RNA/protein comparison using EVO2 / ORTHRUS / ESM3 on the same missense variant set is a compelling setup and makes the paper stand out. It allows a direct cross-modality analysis rather than focusing only on proteins.

3) Thorough empirical analysis

The paper goes beyond a single AUROC table and includes:
	•	per-layer dynamics,
	•	conservation confounder control (partial Spearman + quartile stratification),
	•	metric hierarchy (LID/PR/TSA),
	•	gene-level robustness,
	•	sensitivity to neighborhood size k,
	•	and a feature-combination analysis in the appendix.
This is unusually thorough for a workshop paper and makes the conclusions more convincing.

4) Generally well written and easy to follow

Most sections are clear, especially the introduction, figure framing, and results. The paper communicates the high-level contribution well.


Weaknesses

1) The central geometric hypothesis is under-motivated

The hypothesis (“pathogenic mutations, which violate hard biophysical constraints such as folding stability or active-site accessibility, should concentrate their displacements into low-dimensional subspaces, while benign substitutions scatter across a higher-dimensional space of tolerated variation”) currently reads as post hoc / intuition-driven. The motivation would be stronger with either:
	•	more grounding in prior literature (e.g., related work on manifold geometry), or
	•	a clearer mechanistic argument (why exactly low-dimensional concentration should emerge in these embeddings).

2) Limited dataset scope for a general perspective

The empirical results are on ClinVar missense variants. To support a more general claim, I would like to see at least one additional dataset to test robustness beyond ClinVar’s label and ascertainment biases. A cross-dataset validation would materially improve the contribution.

3) Limited motivation and use-case

The paper is analytically interesting, but the practical use-case is not fully concrete. It would help to clarify:
	•	What are current limitations and challenges. Why it is important to investigate such property.
	•	How geometry could be integrated into downstream pathogenicity pipelines.


Minor comments

1) Method section should include formulas for all primary metrics

Equation for LID is provided, but PR and TSA are only described verbally. Since these are part of the primary results (Section 3.4), I recommend adding explicit formulas/definitions (or a short appendix subsection) for:
	•	PR (participation ratio),
	•	TSA (thresholding / rank criterion used).
This will improve reproducibility and readability.

2) Define terms before first use

Some terms are introduced somewhat abruptly, especially “gap” in Section 3.4. The reader can infer it means pathogenic–benign median difference, but it should be defined explicitly at first use in the main text. Same general comment applies to a few other shorthand terms.

3) Significance in the “57–61% retained” claim

The “57–61% retained” partial-correlation result is interesting, but the difference between 57, 59, and 61 is likely too small to interpret substantively without uncertainty estimates. I suggest adding an uncertainty reporting for these retained percentages if the authors want to discuss modality differences.

4) Add LVD-vs-depth analysis for completeness

The per-layer LID analysis is a highlight of the paper. To complete the story, it would be helpful to include an analogous figure of LVD vs relative depth curves, so readers can directly compare geometric signal vs perturbation magnitude dynamics across layers.

---

### Meta-Review · Area_Chair_wEjd · 2026-02-25

**Recommendation:** Accept (Poster)
**Confidence:** 4

**Metareview:**

Accept.

---

### Decision · Program_Chairs · 2026-03-02

**Decision:**

Accept (Poster)

**Comment:**

Please see the meta-review.